# Preferences for oral and injectable PrEP among qualitative sub-study participants in HPTN 084

**Elizabeth E. Tolley**[1☺*], **Agatha Bula**[2☺], **Miria Chitukuta**[3☺], **Nomhle Ndimande-Khoza**[4☺], **Juliane Etima**[5☺], **Emily Namey**[1‡], **Doreen Kemigisha**[5‡], **Lerato Makhale**[4‡], **Mercy Tsidya**[2‡], **Marie Shoen**[1‡], **Mina C. Hosseinipour**[2,6‡], **Sinead Delany-Moretlwe**[4‡], on behalf of the HPTN 084 study team[¶]

**1** FHI 360, Durham, North Carolina, United States of America, **2** UNC-Project Malawi, Malawi Clinical Research Site, Lilongwe, Malawi, **3** University of Zimbabwe Clinical Trials Research Centre (UZCTRC) - Spilhaus Clinical Research Site, Harare, Zimbabwe, **4** Wits RHI, Faculty of Health Sciences, University of the Witwatersrand, Johannesburg, South Africa, **5** Makarere-Johns Hopkin University Clinical Research Site, Kampala, Uganda, **6** University of North Carolina at Chapel Hill School of Medicine, Chapel Hill, North Carolina, United States of America

☺ These authors contributed equally to this work.
‡ EN, DK, LM, MT, MS, MCH and SD-M also contributed equally to this work.
¶ Membership of the HPTN Study Team is provided in the Acknowledgments.
* btolley@fhi360.org

**Data Availability Statement:** Full qualitative transcripts will not be made publicly available due to the sensitive nature of the information contained therein, and the inability to ensure that a full

## Abstract

### Background

HPTN 084 compared the safety and efficacy of long-acting injectable cabotegravir (CAB) to daily oral TDF/FTC for prevention of HIV-1 in uninfected African women. Like a similar trial in MSM/TGW (HPTN 083), the trial was stopped early for efficacy, expediting the need to consider introduction strategies for different populations. We examine survey and qualitative data from a four-country sub-study to examine oral and injectable PrEP acceptability and considerations for CAB access among African women.

### Methods

Participants completed baseline and follow-up surveys on HIV risk perception, sexual behavior. product acceptability and adherence during the blinded trial. Additionally, up to two in-depth interviews each with 73 sub-study participants explored product use and trial-related experiences, during the blinded and unblinded study periods. Using survey data, we classified participants as: engaged in female sex work (FSW), having multiple non-transactional partners, or monogamous. A study statistician identified participants' assigned study arm. We followed a thematic analysis process to read transcripts, develop a codebook and apply codes in NVivo to transcripts with intermittent intercoder reliability checks; using Excel matrices to explore differences across risk categories and study arms.

transcript is completely de-identified. While direct identifiers and location names were not included in transcripts when they were prepared, there are no standardized procedures and tools for de-identifying sensitive qualitative data and removing indirect identifiers. In addition, qualitative research participants were assured during the informed consent process that their identity would remain confidential. Upon request, approval to access these data may be provided through a Data Use Agreement by FHI 360. Data access requests may be submitted to: Mr. Scott Rose (Clinical Research Operations Manager II, Science Facilitation HPTN Operations) at email "SRose@fhi360.org". Mr. Rose supports operations management for multiple studies conducted through the HPTN, but is not directly involved in implementation and/or analysis and dissemination of the qualitative data for this substudy.

**Funding:** Overall support for the HIV Prevention Trials Network (HPTN) is provided by the National Institute of Allergy and Infectious Diseases (NIAID) National Institute of Allergy and Infectious Diseases (NIAID) (nih.gov), Office of the Director (OD) NIH Office of the Director | National Institutes of Health (NIH), National Institutes of Health (NIH) National Institutes of Health (NIH) | Turning Discovery Into Health, National Institute on Drug Abuse (NIDA) NIDA.NIH.GOV | National Institute on Drug Abuse (NIDA), the National Institute of Mental Health (NIMH) NIMH » Home (nih.gov), and the Eunice Kennedy Shriver National Institute of Child Health and Human Development (NICHD) Homepage | NICHD - Eunice Kennedy Shriver National Institute of Child Health and Human Development (nih.gov) under Award Numbers UM1AI068619 (HPTN Leadership and Operations Center), UM1AI068617 (HPTN Statistical and Data Management Center), and UM1AI068613 (HPTN Laboratory Center). Additional funding was provided by the Bill & Melinda Gates Foundation Bill & Melinda Gates Foundation | Bill & Melinda Gates Foundation., ViiV Healthcare The Global specialist company in HIV Care | ViiV Healthcare, and Gilead Sciences gilead.com provided pharmaceutical support. no additional external funding received.

**Competing interests:** The authors have declared that no competing interests exist.

## Findings

Participants overwhelmingly preferred injections to pills, appreciating the ease, convenience, and privacy of a long-acting formulation. Many participants described challenges with contraceptive and/or study pill adherence, impeded by late night work, unexpected travel, or heavy drinking. Women in the TDF/FTC arm were more likely to describe side effects, compared to those in the CAB arm. Pain also varied widely by study arm. When considering post-trial access to CAB, limited PrEP knowledge, cost and concerns around stigma and poor service quality were potential access barriers.

## Conclusion

Women's desire for privacy and ease of use outweighed injectable concerns, resulting in a strong preference for CAB. Cost and accessibility will need to be addressed by implementation programs.

## Introduction

After more than 35 years of the HIV pandemic, we have made significant progress in reducing HIV-related morbidity and mortality. Yet, in sub-Saharan Africa, women and girls accounted for 63% of all new HIV infections in 2020 [1], with most other new infections among key populations. Since the early days of the pandemic, several new HIV prevention options have been developed and introduced, promising to expand choices for at-risk populations. In 2011, the Food and Drug Administration (FDA) approved the use of TRUVADA, a daily oral combination pill (Tenofovir disoproxil fumarate/Emtricitabine or TDF/FTC) [2] with the World Health Organization's publication of revised guidelines in 2016 and 2015, the potential to use daily oral PrEP as an HIV prevention tool became a reality [3, 4]. Although, a second daily oral product, DESCOVY (emtricitabine and tenofovir alafenamide or TAF/FTC), was approved for pre-exposure prophylactic (PrEP) use in 2016, the indication was for people who are not at risk from vaginal exposure to HIV, limiting women's access to the product [5].

Yet, a decade after the approval and introduction of TDF/FTC, uptake, adherence and persistence of the product is variable. Research and practical experience alike have demonstrated that oral PrEP users face unique challenges when it comes to consistent use of a daily pill [6]. A systematic review of oral PrEP adherence that included 18 prospective randomized controlled trials (RCT) and implementation studies across diverse populations and geographies identified a range of personal, social and product-related factors that influenced adherence, from low-risk perception and/or decision-making power, to social concerns about stigmatization, as well as challenges with product dosing and side effects [7]. Low self-perceived HIV risk [8, 9] and challenges with disclosure of product use to sexual partners and others [10] were identified as reasons for low adherence and sub-optimal PrEP efficacy in several phase 3 RCTs evaluating TDF/FTC among HIV-negative sub-Saharan African women [11–13]. PrEP efficacy was notably higher among women uninfected by HIV in sero-discordant couples [14], presumably because risk perception was higher and adherence to PrEP offered women the ability to preserve both their health and their marriage/partnership [15]. While some research suggests that women using products with known efficacy can and will be more adherent [16], a pooled analysis of 11 post-approval PrEP studies conducted in cisgender women identified varying patterns of adherence over time [17]. Based on a trajectory modelling approach that utilized

available adherence data (including objective measures such as dried blood spots or tenofovir concentration measures, or more subjective measures like pill counts or self-reports) from almost 3000 women, about 40% of women reported consistent use (19%) or consistently high use (22% reported 4–6 pills per week) of PrEP up to 96 weeks of use. An equal proportion of women (39%) across these studies showed decreasing adherence over time, while about 21% reported consistently low adherence [17]. Several smaller demonstration and implementation studies shed further light on these adherence patterns. For example, in a recent retrospective cohort study of 262 female PrEP initiators in Kisumu, Kenya, continuation dropped off significantly, with only 12% documented as having refilled their PrEP pills after 6 months [18, 19]. Similarly, in a PrEP implementation study (HPTN 082) among 451 young women from South Africa and Zimbabwe, the majority (55%) had lapses in PrEP refills over a period of i12 months and only 21–22% were identified as highly adherent at six months [20]. Women randomized to the advanced adherence arm of the study who reported at least one risk factor during a study visit were also more likely to have detectable drug on board, leading researchers to theorize that some young women might try to align their PrEP use with seasons of perceived risk [21]. These social barriers continue to exist for women and make daily product use challenging.

HPTN 084, a multisite, double-blind, randomized Phase 3 trial, compared the safety and efficacy of injectable Cabotegravir (CAB) administered 8-weekly to daily oral TDF/FTC for prevention of HIV-1 in uninfected African women. Initiated in November 2017, the study enrolled 3,224 sexually active women aged 18–45 who were randomized to receive one active (CAB or TDF/FTC) and one placebo product and participated in a 5-week oral run-in before moving into an injection phase [22].

Like a similar trial in cisgender men and transgender women who have sex with men (HPTN 083), the trial was stopped early after demonstrating superiority of CAB over TDF/FTC in preventing HIV. Participants in both trials have been "unblinded" (i.e., informed about which active product they were using), and are currently being followed on the product of their choice (or no product) [22, 23]. The shortened timeline of these two trials has expedited the need to consider introduction strategies for different populations.

The overall goal of this study was to assess acceptability of these two PrEP methods and considerations for CAB access among African women at risk of HIV. To do this, we examine both qualitative and survey data from the initial phase of a four-country sub-study nested within HPTN 084 to assess acceptability of and preferences for injectable versus daily oral PrEP. We consider how women's particular risk contexts and their motivations for trial participation influence product perceptions and experiences.

## Materials and methods

### HPTN 084 overview

As described elsewhere, HPTN 084 was a phase 3, randomized, double-blind, double-dummy trial, conducted in seven sub-Saharan African countries [22]. During the blinded trial phase (from late November 2017-early November 2020), 3224 HIV-uninfected, but PrEP-eligible women (as assessed by using a risk score [24]) aged 18–45 were enrolled and randomized to receive either active CAB or active TDF/FTC, as well as a corresponding daily oral pill or injectable placebo [22]. Participants initially used two daily oral regimens before moving into an injectable phase, during which they received active or placebo injections every eight weeks and also took a daily oral product. On November 4, 2020, while a data safety monitoring review concluded that CAB was superior to TDF/FTC, preparations were made to unmask participants to which active product they were assigned (known as unblinded period) and

subsequently to transition them to an open-label extension study where they could choose to remain or to switch products for an additional 96 weeks. This study is registered with Clinical-Trials.gov, NCT03164564.

## Qualitative sub-study methods

In January 2020, we initiated a prospective qualitative sub-study with a maximum of 104 women from four sites [Lilongwe, Malawi; Johannesburg, South Africa; Kampala, Uganda; Harare, Zimbabwe]. In each sub-study site, from 12–16 women who were actively continuing participation (referred to as "continuing participants" or CPs) were recruited from a list of approximately 30 randomly selected enrolled participants representing early (2018), mid-trial (2019) or later entry (2020) into the trial. CPs received up to three in-depth interviews (IDIs), spanning participation in the blinded trial, an unblinded phase during which women continued to use their assigned active product, and an open-label phase choice phase during which they could remain on their assigned product or switch. In addition, sites recruited up to 10 "special cases" (SCs) for up to two serial interviews. SCs were invited to participate in the qualitative sub-study if they became pregnant, were put on product hold, or sero-converted. The first sub-study IDI was conducted on August 26, 2020; the last IDI of a special case for this paper was conducted on November 22, 2022. The study protocol, including its qualitative component, was approved by the Institutional Review Boards at each corresponding study site. Written consent was obtained from all participants.

## Data collection

HPTN 084 participants completed a computer-assisted structured interviews (CASI) at baseline and multiple follow-up visits; baseline questions assessed participants' relationship and household contexts, HIV risk perceptions, sexual and other risk behaviors, including having been paid for sex in the last month or considering oneself a sex worker, and initial perceptions of injectable and daily oral PrEP likes and dislikes. At multiple follow-up visits, they reported on acceptability and use of injectable and daily oral regimens, as well as perceived ability to adhere to daily oral pills. In addition, qualitative sub-study participants took part in up to three in-depth interviews. The initial interview inquired about reasons for trial participation, and initial experiences using study products. Follow up interviews examined in-depth women's experiences using injectable and daily oral products, acceptability of and preferences for various PrEP options, as well whether or how they might want to access and use PrEP once the trial had concluded. All IDIs were conducted by same-sex interviewers in local language at study sites and lasted approximately one hour. Interviewers were trained in qualitative data collection techniques and interview guide content.

## Data analysis

Quantitative tables were prepared by a statistician assigned to the HPTN 084 clinical trial. Qualitative analyses were led by a cross-site team of 10 researchers training in qualitative thematic analysis. Qualitative analyses presented in this paper focus on the first and second IDIs of CPs and SCs, both of which were conducted prior to participants' ability to switch or remain on their current active product, i.e., during the blinded and unblinded periods of the study. The research teams followed a four-step process to 1) read transcripts for emerging themes (e.g., Sexual History, Product-related Acceptability, Adherence, Pregnancy, PrEP Use, and Clinical Trial Experiences); 2) developed a codebook and applied codes in NVivo version 12 to transcripts with intermittent virtual meetings to conduct interrater reliability checks; 3) developed memos identifying sub-themes and illustrative contexts for main codes; and 4)

summarized information about each participant, including personal characteristics, site and assigned study arm, in Excel matrices to explore how product-related experiences might differ across these attributes. We also explored differences across risk categories related to product acceptability and other themes [25]. We used a combination of CASI and IDI data to develop risk profiles. Specifically, we constructed four categories of risk. Based on the baseline CASI responses, two categories included: 1) women who reported sex work; 2) women who reported having been paid for sex in the past month, but did not report sex work. Among women who did not report transactional sex or sex work in the baseline CASI, we constructed two additional categories based on information provided in their first qualitative interview, including: 3) women who described multiple sexual relationships; and 4) women who reported being monogamous. In addition to examining any differences across risk categories, we compared product-related experiences by assigned study arm.

## Findings

A total of 76 participants were recruited into the qualitative sub-study. Sub-study participants' risk contexts varied across sites, and their motivations to join HPTN 084 were multiple. Over a third of women reported being monogamous, either married, cohabiting or in committed relationships (Table 1). Others described multiple, sequential, or concurrent partners–some relationships were more transactional, and about one-fifth of participants described

Table 1. Socio-demographic characteristics of sub-study participants, by country.

| | All | Malawi | South Africa | Uganda | Zimbabwe |
|---|---|---|---|---|---|
| | (n = 76) | (n = 20) | (n = 20) | (n = 17) | (n = 19) |
| Age (mean) | 24.6 | 23.9 | 23.1 | 24.7 | 27.0 |
| **Marital status** | % | % | % | % | % |
| • Married/civil union/legal | 16 | 10 | 0 | 0 | 53 |
| • Living with primary partner | 7 | 10 | 5 | 6 | 5 |
| • Has primary partner, not living together | 32 | 10 | 95 | 12 | 5 |
| • Single/divorced/widowed | 46 | 70 | 0 | 82 | 37 |
| **Current employment status** | % | % | % | % | % |
| • Full time | 12 | 0 | 5 | 47 | 0 |
| • Part time | 11 | 5 | 10 | 6 | 21 |
| • Not employed | 78 | 95 | 85 | 47 | 79 |
| **Highest education level** | % | % | % | % | % |
| • Primary | 16 | 40 | 0 | 18 | 5 |
| • Secondary | 76 | 60 | 70 | 82 | 95 |
| • Technical training | 1 | 0 | 5 | 0 | 0 |
| • College/university or higher | 7 | 0 | 25 | 0 | 0 |
| **Nights sleep at home every week** (median, Q1,Q3) | 6.0 (4,7) | 7.0 (4.7) | 5.0 (4,6) | 4.5 (3,7) | 7.0 (6,7) |
| **Who living with\*:** | % | % | % | % | % |
| • Alone | 5 | 0 | 5 | 12 | |
| • Partner | 20 | 0 | 15 | 12 | 5 |
| • Parents | 38 | 50 | 55 | 24 | 53 |
| • Siblings | 24 | 30 | 25 | 29 | 21 |
| • Own children | 32 | 55 | 10 | 6 | 11 |
| • Roommates | 12 | 10 | 10 | 29 | 053 |

\*Multiple responses possible. Sum of responses may exceed 100%.

**Table 2. Risk behaviors of sub-study participants.**

| | All | Malawi | South Africa | Uganda | Zimbabwe |
|---|---|---|---|---|---|
| | (n = 76) | (n = 20) | (n = 20) | (n = 17) | (n = 19) |
| **How at risk of getting HIV?** | % | % | % | % | % |
| Not at all | 8 | 10 | 5 | 0 | 16 |
| A little | 25 | 45 | 25 | 24 | 5 |
| A lot | 53 | 35 | 25 | 76 | 79 |
| Prefer not to answer | 4 | 10 | 5 | 0 | 0 |
| *Missing* | 11 | 0 | 40 | 0 | 0 |
| **Risk from own behaviors?** | % | % | % | % | % |
| Not at all | 21 | 30 | 20 | 6 | 26 |
| A little | 24 | 30 | 25 | 24 | 16 |
| A lot | 39 | 30 | 10 | 65 | 58 |
| Prefer not to answer | 5 | 10 | 5 | 6 | 0 |
| *Missing* | 11 | 0 | 40 | 0 | 0 |
| **Risk from partners' behaviors** | % | % | % | % | % |
| Not at all | 8 | 15 | 5 | 0 | 11 |
| A little | 17 | 15 | 20 | 24 | 11 |
| A lot | 64 | 70 | 35 | 76 | 79 |
| Prefer not to answer | 0 | 0 | 0 | 0 | 0 |
| *Missing* | 11 | 0 | 40 | 0 | 0 |
| **Ever been paid for sex, last month** | % | % | % | % | % |
| Yes | 41 | 55 | 15 | 65 | 32 |
| No | 58 | 45 | 80 | 35 | 68 |
| Prefer not to answer | 1 | 0 | 5 | 0 | 0 |
| **Identify as a sex worker** | % | % | % | % | % |
| Yes | 25 | 30 | 10 | 41 | 21 |
| No | 71 | 65 | 85 | 59 | 74 |
| Prefer not to answer | 4 | 5 | 5 | 0 | 5 |

themselves as sex workers (Table 2). Across contexts, women joined the study to gain access to HIV prevention products, either due to concern about their own risk behaviors or because they suspected their partners of having other sexual partners. Women also sought access to trial-related benefits including free long-acting contraception, HIV testing or testing for other health conditions, or desired financial reimbursements or new experiences.

Our qualitative sub-study sample varied in distinct ways across the four sites. Participants from South Africa were the youngest group, on average, while women from Zimbabwe were oldest. While most South African participants had a primary partner, only 1 (5%) lived with her partner. In contrast, 10 of 19 (53%) Zimbabwe participants lived with their partner. Most participants from Malawi and Uganda were single, divorced or widowed and thus without a primary or main partner. Except for Uganda, few participants in the sub-study reported full or part-time jobs; some South African participants were students at university or technical colleges. Finally, while about half of sub-study participants in Malawi and Zimbabwe lived with their own children, this was less often the case for South African and Ugandan participants (Table 1).

Over 40% (31 of 76) of our qualitative sample reported at baseline having been paid for sex in the month prior to baseline; one-quarter of the sample identified as a sex worker (Table 2). Based on additional qualitative data, of those who didn't report baseline transactional sex, 19

**Table 3. Risk category distribution of sub-study participants, by study arm.**

| Risk Category | TDF/FTC Arm | CAB LA Arm | Total |
|---|---|---|---|
| FSW | 9 | 10 | 19 |
| Transactional sex | 8 | 4 | 12 |
| Multiple partners | 11 | 8 | 19 |
| Monogamous | 11 | 15 | 26 |

women reported having multiple partners while 26 reported being monogamous. Distribution of risk by study arm was similar (Table 3).

To varying degrees, participants' perceptions of and experiences using injectable and oral PrEP options differed by these risk categories. Furthermore, when asked about whether or how they might want to use PrEP options once the clinical trial had ended, women's perspectives on PrEP use beyond their trial participation, and potential barriers to accessing PrEP outside of the trial, reflected their risk contexts.

## Initial expectations about injectable and oral pill study products

At enrollment, women had certain expectations about what they might like from oral or injectable PrEP (Fig 1a and 1b). For example, sub-study participants most liked the idea that

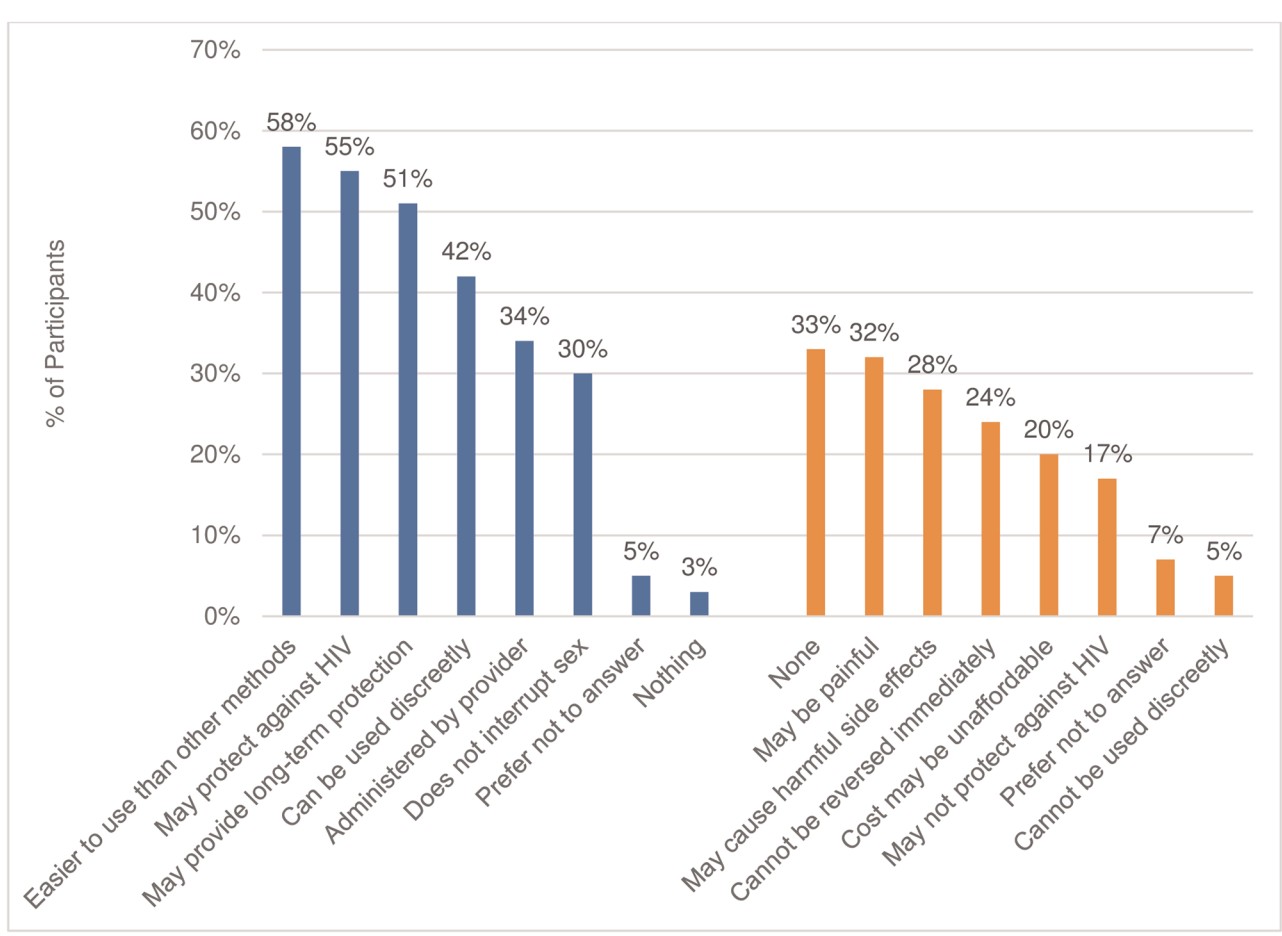

**Fig 1. Injectable PrEP likes and dislikes at baseline (n = 76) what do you think you might like about an injectable PrEP method? What concerns do you have about an injectable HIV prevention method?.**

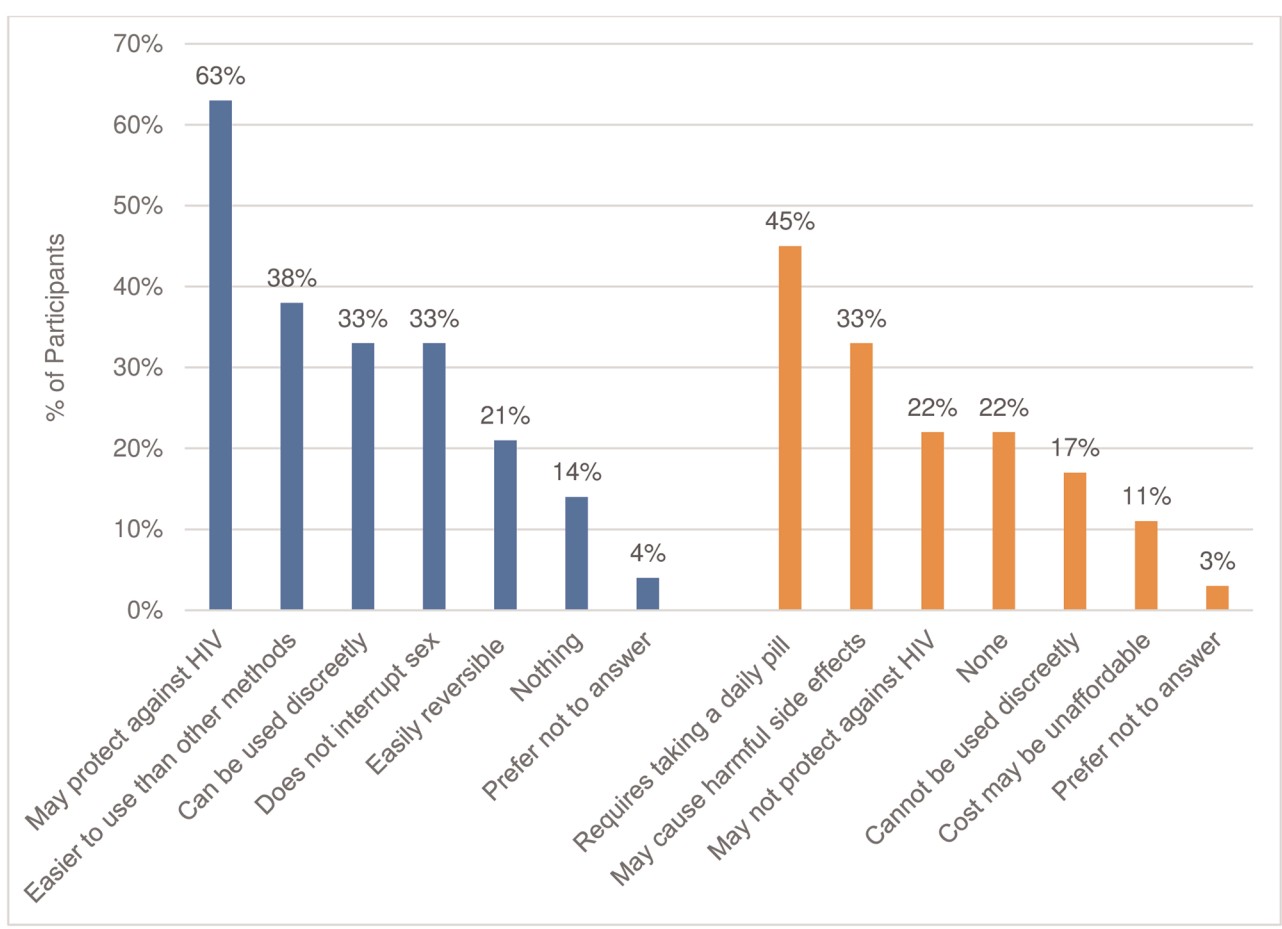

**Fig 2. Oral PrEP likes and dislikes at baseline (n = 76) what do you think you might like about an oral PrEP method? What concerns do you have about an oral HIV prevention method?.**

study products, whether oral pills or the injectable, would protect them from HIV. While slightly more women attributed HIV protection to oral compared to injectable PrEP–likely due to its known efficacy, their expectations of injectable versus oral PrEP were more favorable in terms of longer-term protection, discreetness of use, and ease of administration. More than one-third of sub-study participants had no concerns about injectable PrEP compared to less than one-fourth of participants with no concern about oral PrEP (Fig 2).

## Injectable product experiences

By far, women appreciated the ease and convenience of a long-acting formulation compared to a daily oral regimen. Injectable PrEP did not require daily remembering and fit better into women's lifestyles, especially for women who traveled or had unpredictable work. For women who identified their own behavior as putting them at risk of HIV, the longer duration of the injectable was perceived as more effective protection (even if, ultimately, they had been randomized to the TDF/FTC arm).

> *I joined this study because when my marriage ended, I was promiscuous and when I heard that there is a study where there is a product that prevents HIV, I decided to join. During that*

*time, with the nature of my work, I thought I was being protected because I was having unprotected sex with multiple sexual partners. Sure. So, there was a certain month when I was reckless because I went away for a long time, and I stopped taking my pills. . .. I had more confidence in the injection and although I missed the oral pills, I had the feeling that the injection would protect me because it is long acting.*

*(Malawi, 29, divorced, sex worker, TDF/FTC arm)*

*As I mentioned before, it's just a matter of getting my jab and going home. I do not have to carry anything else with me. This makes the injection different from pills which might get lost on the way, or which I might forget to take. I might travel to some place and forget to carry the pills with me. With the injection, I have no such pressures. After getting injected, it's done.*

*(Zimbabwe, 25, married, monogamous, CAB arm)*

Regardless of risk category, women liked the injectable's privacy from husbands, boyfriends, sexual clients or just "nosey people". Most women in the monogamous risk category had disclosed study participation to either a partner, a family member, or multiple people. Women in other risk categories seemed to carefully weigh what they could disclose–whether study participation or product use–and to whom. Disclosure was at times necessary when a household member inadvertently found a woman's study pills.

*Yeah, taking pills is very hard hey, especially if you are someone like me who is never home, no, it's quite difficult. MMM. You have to pack your bags and be somewhere, maybe with your friends, [be]cause a few of my friends know about it, but others they don't, I don't feel like they should know.*

*(South Africa, 23, not married, transactional sex, CAB arm)*

The main disadvantage of injectable PrEP was pain. Almost all women described some level of pain, but descriptions varied widely by study arm. For example, only a few participants randomized to the CAB arm, but many more in the TDF/FTC arm (5 versus 20), suggested that injectable pain was easily tolerated. These women described the level of pain as normal, lasting just a few minutes, or disappearing before they got home from the clinic. Indeed, a few women wondered whether the lack of pain signified that they were on the placebo.

*HOW PAINFUL OR PAINLESS WOULD YOU DESCRIBE THIS INJECTABLE? I don't feel anything when they inject me. But when I wake up the following day, I feel the pain, but it's bearable. But the moment you walk out of there, you can feel that there is something, but it's bearable.*

*(South Africa, 27, not married, multiple partners, CAB arm)*

*BUT YOU LATER NORMALIZED. WHAT ELSE DID YOU EXPERIENCE APART FROM THAT? Nothing, nothing has happened ever since, because when I started using the injection nothing happens to me. It has never caused any pain, nothing really. I used to fear it when they were injecting me. ARE YOU STILL FEARING IT, FOR THE PERIOD YOU HAVE SPENT IN THE STUDY? I got used to it. HOW COME YOU GOT USED TO IT? At first, I thought that it is very painful, but it was not.*

*(Uganda, 22, not married, monogamous, TDF/FTC arm)*

In contrast, many women described moderate to significant levels of pain, with some differences by arm. Women who received the placebo injections were more likely to describe sharp– but short-duration bouts of pain (e.g., minutes or hours), while those in the CAB arm described pain that might last for days or be accompanied by other site reactions.

*HOW EASY OR DIFFICULT WAS IT TO GET BOTH THE TWO-MONTHLY INJECTION AND TAKE A DAILY PILL? It was easy. The challenge was the injection, because after a shot, you could really feel that you have been injected. THE PAIN. . . Yes, the pain. You could feel it moving inside your body and after the shot, you could hardly walk because of the pain and numbness. . . . IF YOU WERE ASKED TO TAKE ONE PRODUCT, WHICH ONE WOULD YOU HAVE PREFERRED? WHY? The pills. . . .. WHAT IS IT THAT YOU DID NOT LIKE ABOUT THE INJECTION? What didn't I like? YES. . . The pain. After injecting, it takes some minutes before the pain stops.*

*(Malawi, 25, divorced, monogamous, TDF/FTC arm)*

*OKAY AND WHAT DID YOU DISLIKE ABOUT IT (injection)? The first few injections were so hard, painful. You could feel it. Not the injection, not the needle, but the medicine as it's going into your system, you'd feel it. And sometimes I'd stay for two weeks with a swollen bum that I cannot even touch [inhales]. Like even when sitting, yooooooh, when sleeping when you are turning you have to wake up like, [inhales sharply] "oooh, I turned on the wrong side!" MM. That was painful. BUT WHAT DID YOU TAKE TO MAKE THE SWELLING GO AWAY? . . . They gave me pain killers, mm hmm (yes). But even, like in the beginning, even when I was taking pain killers, I would have to take them for like three or four days for it to go down.*

*(South Africa, 38, not married, multiple partners, CAB arm)*

About three-fourths of participants had ever used injectable contraception. Oftentimes, they would compare the level of pain they experienced with study injections to injectable contraception. Despite these direct comparisons, however, both active and placebo study injections were described as more–or less painful than the contraceptive injection. In addition, some women attributed the level of pain to the skill of the clinic staff.

*The Depo injection is more painful. . . Aah as for this one (referring to the study injection), it is not painful. I would not know if it is the one which is not painful (e.g., placebo), or it's the people who make it painless.*

*(Zimbabwe, 24, divorced, monogamous, TDF/FTC arm)*

A few women (10 of 76) complained of other side effects from injections, including injection site swelling and itchiness, leg cramps, or post-injection dizziness. Several worried about potential longer-term health effects, especially on an unborn baby.

## Oral product experiences

About one-third of participants found oral pills easy to take and mentioned low side effects. But, more than half of women worried about forgetting to take oral pills. Some referenced prior mishaps, including unintended pregnancies, when taking oral contraceptive pills. Pill attributes–size, taste and smell were disliked by some.

*HOW DID IT APPEAR TO YOU WHEN YOU FIRST SAW IT? I thought I would not be able to swallow it. I have always detested pills. That is why I never took family planning tablets because I detest pills.*

*[Laughs] (Zimbabwe, 33, divorced, monogamous, TDF/FTC arm)*

Additionally, more than half of participants (41 of 76) described encountering some problems with study pill adherence. For some, adherence was challenging early in the study as they figured out the best timing to take pills. Forgetting to take their pills due to late night work, when travel unexpectedly came up, or after heavy drinking was mentioned mostly by women who acknowledged sex work or engaged in transactional sex.

*I had just started participating, I would forget. HOW MANY DAYS DID YOU FORGET? I won't lie to you. They were a bit many. I think about a week but of course not consecutively, you would forget today but you take it the following day, like that.*

*(Uganda, 22, not married, monogamous, TDF/FTC arm)*

Less frequently (31 of 76), women listed an array of side effects they attributed to oral pill use. Side effects were more often described by women randomized to the TDF/FTC arm (20 versus 11) and included: hunger pains, dizziness, nausea, headache, diarrhea, sleepiness, or sweating. Participants often stated that these side effects subsided over time.

*At the start of the study, I got bad effects from the tablets before my body got used to them [tablets]. But now my body is used to these tablets, and I do not have any problems with them. I take them well because I take them at 7.00p.m. HMM. WHICH START ARE YOU TALK-ING ABOUT WHEN YOU GOT SIDE EFFECTS? At the very start of the study when we were taking two tablets daily. I got side effects of nausea, diarrhea and had abdominal pain all the time, but now my body got used and I no longer get those side effects.*

*(Uganda, 34, married, multiple partners, TDF/FTC arm)*

## PrEP preferences

Regardless of study arm, about 80% of women who expressed an opinion about product preference preferred injections to daily pills (41 versus 9 of 50); they often explained that the injection would be harder to forget to take and meant fewer clinic visits.

Nevertheless, some women acknowledged that preference was personal–some disliked swallowing pills, while others were afraid of injections. Women in higher risk categories (FSW and transactional sex) were more likely to say they would find either product acceptable, and were more likely than other groups to include use of condoms along with a PrEP option.

*The injection is painful, but I will choose it. MAY YOU TELL ME WHY? [Be]cause it's a one-time thing. You just have to remember the date, but after that it's over. AND IF MAYBE WE FOUND THAT THE PILL WORKS BETTER, WOULD YOU STILL PICK THE INJECTION? No, I'll go for the pill. A pill is not bad, hey, the part of having to remem-ber every day.*

*(South Africa, 28, not married, multiple partners, CAB arm)*

*Aah, all things have advantages and disadvantages. . . .For those who forget the pills, if they have the injection, it will be easier for them. But for those who find the injection painful, they would rather have the pills then it also becomes easy for them. So, a person will make a choice.*

*(Zimbabwe, 23, married, monogamous, CAB arm)*

## PrEP access beyond the trial

At the time of these interviews, participants' knowledge about how to access PrEP outside of the clinical trial was generally low. While a few women knew of others who were taking either antiretrovirals for treatment of HIV or using oral PrEP, most participants first learned about PrEP when they were recruited for the study. Women who provided information about access tended to refer generically to clinics, while several specifically mentioned clinics for sex workers. A few participants believed that PrEP might be available at pharmacies or the hospital, although few women had previously sought PrEP or knew someone who had.

More than half of women in the FSW category desired continuing use of PrEP–preferably injectable PrEP, once the trial had ended. They reported seeking oral or injectable PrEP, if available, at a nearby pharmacy or local clinic, and being willing to pay for PrEP, if needed.

*SOME WOMEN WHO MAY NOT BE ABLE TO ACCESS THE INJECTABLE PREP BUT THEY ARE WILLING TO USE IT. HOW DO YOU THINK THIS CAN AFFECT THEIR WILLINGNESS? If it can be available for sale, those women who can be willing to buy, they can buy it because they want to protect themselves. HOW MUCH DO YOU THINK THEY CAN BE ABLE TO PAY FOR THE INJECTABLE PREP? K2000 OR K3000. I can even buy if it can cost K5000 (about $5.00) for the sake of protecting myself, because I cannot buy life with the same money.*

*(Malawi, 28, divorced, female sex worker, CAB arm)*

Fewer women in other categories were clear about their continued PrEP use. Cost was often mentioned as a potential barrier to PrEP access.

*YOU MENTIONED THAT YOU MET SOMEONE WHO HAD PREP, DIDN'T YOU? Yes. THAT MEANS IT'S ACTUALLY AVAILABLE IN THE PHARMACIES. WOULD YOU WANT TO USE IT AS AN HIV PREVENTION METHOD? . . . If it's available and affordable, I might use it but if it's expensive that would be problematic. OK. WHAT PRICE WOULD YOU FIND UNAFFORDABLE? If it costs US $5 or US $10, I might not afford it.*

*(Zimbabwe, 43, married, transactional sex, CAB arm)*

For many, future use of PrEP was contingent on being able to obtain it for free through the study clinic or a public sector clinic providing free or subsidized services. However, even when affordable, some women worried about having to access PrEP through public clinics due to perceived stigma from both providers and other clients, as well as poor services.

*LET'S SAY THE 084 STUDY COMES TO AN END, YOU WON'T BE ACCESSING [PrEP] PILLS BUT PrEP IS AVAILBLE AT THAT CLINIC, ARE YOU COMFORTABLE TO GO THERE TO ACCESS PrEP? Yes. WHAT WOULD HAVE CHANGED NOW, SINCE BEFORE YOU SAID YOU WOULDN'T ACCESS PrEP FROM THERE? Nothing would have changed, it's just that you would be ashamed of being seen at the clinic for sex workers, so I*

*will also be labeled a sex worker. You may meet someone from your neighborhood who is a sex worker, so when she meets you in the neighborhood now, she will say, "Ah isn't it we see you at the clinic for sex workers, so you are also a sex worker." So that is what would discourage other people from going there.*

*(Zimbabwe, 24, divorced, monogamous, TDF/FTC arm)*

*HMM, DO YOU KNOW ANYONE WHO IS GETTING PREP FROM THE [LOCAL] CLINIC? In our public clinic? YEAH. No.. . .Yeah, even. . . isn't that. . . what is this study that gives PrEP only? It's also [HPTN] 084, but it's finished, they had to go to the public [clinic]. I know people who participated [in the trial], but they are scared to go to the public [clinic]. HMM. Yeah, I think it has to do with that [word] "public". Those words, you know [laughs]. . .WHAT ARE THEY SCARED OF? No, I don't know maybe going inside and asking where one can get PrEP, in those [consultation] rooms, where should you go, like such things that you would ask. Like, it even seems difficult to ask where family planning is done, yoh. . .. [LAUGHS]. That look. HMM. And you know public clinic, there are people from your neighborhood, like it's very difficult and you become scared. [People will] say, "What are you going to do there?"*

*(South Africa, 23, not married, transactional sex, CAB arm)*

## Discussion

Overall, women in this sub-study of the HPTN 084 trial found injectable PrEP to be highly acceptable and preferable to daily oral PrEP. Our qualitative data support findings from previous clinical trials and observational studies that suggest women's preference for injectable over daily oral PrEP [26–30] was in large part because they are discreet, longer-acting and thus easier to adhere to than oral PrEP. In studies across geographic settings and population groups, participants delayed, declined or poorly adhered to oral PrEP out of fear that they might be stigmatized as promiscuous and/or taking HIV treatment drugs, or concerns that accidental disclosure of product use to partners or other family members could lead to violence [31–35]. Similarly, poor adherence and/or discontinued use of oral PrEP have been attributed to challenges remembering to plan for and adhere to product use when social routines are disrupted [31, 36].

Several concerns about injectable PrEP emerged, however, with the main one being pain. In the qualitative sub-study, descriptions of pain varied widely–but tended to align with the type of product being used. Reflecting findings from main study, similar levels of grade 2 or higher adverse events were reported across study arms, with the exception of injection site reactions (ISR), reported by 12.6% of participants in the CAB arm versus 1.6% on TDF/FTC. Reports of ISRs diminished over time and were not associated with discontinuation of study products [22]. As in previous studies, variable levels of pain were a common challenge of the injectable [26, 30], and in a Phase 2 RCT of CAB (HPTN 077) was significantly higher in the CAB versus the placebo arm [27]. Nevertheless, participants' reports of future interest in using CAB did not vary by treatment arm or level of pain [27]. In a recently published systematic review of values and preferences for injectable PrEP, studies highlighted both the perceived benefit of using a product that could better fit individual lifestyles and concerns about injectable site pain, location of injections and logistical challenges [37]. It remains unclear how much of a disincentive fear of or experience with pain will be for CAB uptake in PrEP

programs. It is possible that willingness to tolerate pain may depend on the degree to which women see PrEP as beneficial–a prevention mentality.

As with contraception, women recognized the potential for choice among a range of alternatives for HIV prevention to suit their different lifestyles. Women's familiarity with injectable contraception and their dislike of oral PrEP attributes (e.g., daily dosing, oral administration, or side effects) were significantly associated with preference for injectable over daily oral PrEP in a study of 394 pregnant or breastfeeding women (PBFW) from SA and Kenya [29]. Similarly, in a prospective cohort study of 425 women in KwaZulu-Natal, South Africa, women expressed preference of HIV prevention method, whether injectable, implant or oral pill, was significantly associated with both current and ever use of a similar contraceptive format [38].

In our study, women in higher-risk categories were more likely to mention effectiveness as influencing interest in using any PrEP product outside of the trial. While most women said that they would prefer to use a long acting injectable, some said they would prefer to move to daily oral pills to avoid future injections, which they found as more painful than contraceptive injections. Some sub-study participants who became pregnant during the blinded portion of the study also expressed a preference to use TDF/FTC during their pregnancy. This preference may reflect the relative lack of safety data on use of CAB during pregnancy, as described in the informed consent form for the blinded trial. Other factors considered by women across risk categories included convenience, side effects, access, and cost.

No data exist yet to indicate how well women will adhere to and/or persist on CAB outside the trial setting. As multiple researchers acknowledge, beyond preference and choice, contextual factors related to women's social networks, characteristics of the healthcare system and broader cultural norms will play important roles in this. While daily oral pills were hard to hide and generated concerns about being stigmatized, it's not clear whether the process of obtaining CAB injections might also subject women to stigma. This could depend on where women must go to obtain it. In our study, it was surprising how few women knew of any facility where they might obtain existing PrEP options; others associated PrEP services with clinics serving sex workers or other high-risk groups. In a survey of almost 300 clients from a diverse set of South African clinics serving SW and MSM populations, most clients had heard of PrEP and two-thirds were currently (38%) or had previously (29%) used PrEP. Almost half of those who had never used PrEP had not been offered it; concerns about side effects, stigma, and ability to adhere also factored into non-use [33]. Lessons from the early PrEP clinical trials and implementation studies emphasized the need for positively framed awareness and demand creation efforts and integrated service delivery strategies as ways to reduce stigma and logistical barriers, thus better reaching women in African settings [6, 39]. More recently, PrEP programs in South Africa are offered to all persons at risk of HIV acquisition, with the goal of normalizing PrEP use and increasing PrEP accessibility [40].

Finally, while some participants expressed a commitment to seeking and paying for injectable PrEP once the trial was concluded, and if available, many participants acknowledged that future use of CAB would likely depend on whether they could obtain it at low or no cost. In recent discussions between the manufacturer and international organizations, price estimates have ranged from $16–270 per shot [41]. Achieving these lower per-unit prices will involve expanding demand and uptake of CAB beyond those who are already accessing oral PrEP, and across multiple countries in the region. This will require careful coordination between governments, donors, product suppliers and others [42]. At the country level, program managers will need to consider the costs and complex implementation strategies necessary to reach the diverse range of women and men who might benefit from PrEP use [43]. Decisions about where to deliver injectable PrEP, whether through services for high-risk "key" populations, routine HIV testing and treatment programs, antenatal or family planning clinics,

or an expanded set of options like community-based services and pharmacies, will likely determine the pace and direction of scale-up [36, 44].

Several strengths of this qualitative sub-study relate to its diverse sample of women in terms of geographic, socio-demographic and risk contexts, as well as an in-depth and prospective exploration of daily oral versus injectable product experiences, acceptability, and preferences. Our ability to examine differences in acceptability and preference by assigned study arm, even as participants remained blinded to their assigned active product, is also a strength. However, we also note several limitations. First, despite our diverse sample, women in this trial–as in all clinical trials, are likely to differ in important ways from women who may ultimately choose to use PrEP. They are reimbursed for study visits and receive products and services for free. They may perceive the services they receive as superior to those they might routinely get outside of a trial, which could influence how they assess acceptability of either product. Finally, our analysis does not include interviews with sub-study participants that were conducted after they were able to actively choose to transition onto or off CAB, so preference-related information is still hypothetical. Nevertheless, we believe these findings highlight the diverse contexts in which women may desire use of a long-acting injectable and provide insights into demand creation and delivery strategies that will be needed to ensure access.

## Conclusion

Women's desire for privacy and ease of use outweighed other injectable concerns, resulting in a strong preference for injectable PrEP. Concerns about cost and accessibility will need to be addressed by implementation programs.

## Supporting information

**S1 Table. Injectable PrEP likes and dislikes at baseline.**
(DOCX)

**S2 Table. Oral PrEP likes and dislikes at baseline.**
(DOCX)

**S1 File. Inclusivity in global research.**
(DOCX)

## Acknowledgments

We wish to acknowledge the contribution of the participants, their families and communities, and the community advisory boards at the four sites participating in the qualitative sub-study. We acknowledge the contribution of the HPTN 084 study team. Members include: Scott Rose, Jennifer Farrior, Jill Stanton, Molly Dyer, Rhonda White, Marcus Bolton Bryan, Jonathan Lucas, Abraham Johnson, Jontraye Davis, Kevin Bokoch, Laura Smith, Nirupama Sista, Aida Asmelash, Yaw Agyei, Myron S. Cohen, Stephanie Orme, Lynda Emel, Susan Eshleman, Mark Marzinke, Estelle M. Piwowar-Manning, Brett Hanscom, Alex Rinehart, Craig W. Hendrix, James Rooney, Sybil Hosek, Katherine Shin, Lydia Soto-Torres, Kimberly Smith, Lut Van Damme. We would like to thank the SCHARP data team for their assistance with preparation of quantitative tables for this sub-study!

## Author Contributions

**Conceptualization:** Elizabeth E. Tolley, Mina C. Hosseinipour, Sinead Delany- Moretlwe.

**Data curation:** Agatha Bula, Miria Chitukuta, Nomhle Ndimande-Khoza, Juliane Etima, Doreen Kemigisha.

**Formal analysis:** Agatha Bula, Miria Chitukuta, Nomhle Ndimande-Khoza, Juliane Etima, Doreen Kemigisha, Lerato Makhale, Mercy Tsidya.

**Investigation:** Elizabeth E. Tolley, Doreen Kemigisha, Lerato Makhale, Mercy Tsidya.

**Methodology:** Elizabeth E. Tolley.

**Project administration:** Agatha Bula, Miria Chitukuta, Nomhle Ndimande-Khoza, Juliane Etima, Emily Namey, Marie Shoen.

**Supervision:** Elizabeth E. Tolley, Agatha Bula, Miria Chitukuta, Nomhle Ndimande-Khoza, Juliane Etima, Emily Namey.

**Validation:** Elizabeth E. Tolley.

**Visualization:** Marie Shoen.

**Writing – original draft:** Elizabeth E. Tolley.

**Writing – review & editing:** Elizabeth E. Tolley, Agatha Bula, Miria Chitukuta, Nomhle Ndimande-Khoza, Juliane Etima, Emily Namey, Doreen Kemigisha, Lerato Makhale, Mercy Tsidya, Marie Shoen, Mina C. Hosseinipour, Sinead Delany- Moretlwe.

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
