## [Decision Letter · Decision Letter 0]

23 May 2024

PONE-D-23-42740Preferences for Oral and Injectable PrEP among qualitative sub-study participants in HPTN 084PLOS ONE

Dear Dr. Tolley,

Thank you for submitting your manuscript to PLOS ONE. After careful consideration, we feel that it has merit but does not fully meet PLOS ONE’s publication criteria as it currently stands. Therefore, we invite you to submit a revised version of the manuscript that addresses the points raised during the review process.

We look forward to receiving your revised manuscript.

Kind regards,

Tinashe Mudzviti, MPhil(MD)

Academic Editor

PLOS ONE

Journal Requirements:

3. Please ensure you have included the registration number for the clinical trial referenced in the manuscript.

Overall support for the HIV Prevention Trials Network (HPTN) is provided by the National Institute of Allergy and Infectious Diseases (NIAID) National Institute of Allergy and Infectious Diseases (NIAID) (nih.gov), Office of the Director (OD) NIH Office of the Director | National Institutes of Health (NIH), National Institutes of Health (NIH) National Institutes of Health (NIH) | Turning Discovery Into Health, National Institute on Drug Abuse (NIDA) NIDA.NIH.GOV | National Institute on Drug Abuse (NIDA), the National Institute of Mental Health (NIMH) NIMH » Home (nih.gov), and the Eunice Kennedy Shriver National Institute of Child Health and Human Development (NICHD) Homepage | NICHD - Eunice Kennedy Shriver National Institute of Child Health and Human Development (nih.gov) under Award Numbers UM1AI068619 (HPTN Leadership and Operations Center), UM1AI068617 (HPTN Statistical and Data Management Center), and UM1AI068613 (HPTN Laboratory Center). Additional funding was provided by the Bill & Melinda Gates Foundation Bill & Melinda Gates Foundation | Bill & Melinda Gates Foundation., ViiV Healthcare The Global specialist company in HIV Care | ViiV Healthcare, and Gilead Sciences gilead.com provided pharmaceutical support.

This study was made possible through funding support from the National Institute of Allergy and Infectious Diseases, Office of the Director, National Institutes of Health (NIH), National Institute on Drug Abuse, and the National Institute of Mental Health. The content is solely the responsibility of the authors and does not necessarily represent the official views of the NIH. Additional funding was provided by the Bill & Melinda Gates Foundation, Gilead Sciences, and ViiV Healthcare. Pharmaceutical support was provided by ViiV Healthcare and Gilead Sciences. We wish to acknowledge the contribution of the participants, their families and communities, and the community advisory boards at the four sites participating in the qualitative sub-study. We acknowledge the contribution of the HPTN 084 study team.  Shout out the SCHARP data team!

Overall support for the HIV Prevention Trials Network (HPTN) is provided by the National Institute of Allergy and Infectious Diseases (NIAID) National Institute of Allergy and Infectious Diseases (NIAID) (nih.gov), Office of the Director (OD) NIH Office of the Director | National Institutes of Health (NIH), National Institutes of Health (NIH) National Institutes of Health (NIH) | Turning Discovery Into Health, National Institute on Drug Abuse (NIDA) NIDA.NIH.GOV | National Institute on Drug Abuse (NIDA), the National Institute of Mental Health (NIMH) NIMH » Home (nih.gov), and the Eunice Kennedy Shriver National Institute of Child Health and Human Development (NICHD) Homepage | NICHD - Eunice Kennedy Shriver National Institute of Child Health and Human Development (nih.gov) under Award Numbers UM1AI068619 (HPTN Leadership and Operations Center), UM1AI068617 (HPTN Statistical and Data Management Center), and UM1AI068613 (HPTN Laboratory Center). Additional funding was provided by the Bill & Melinda Gates Foundation Bill & Melinda Gates Foundation | Bill & Melinda Gates Foundation., ViiV Healthcare The Global specialist company in HIV Care | ViiV Healthcare, and Gilead Sciences gilead.com provided pharmaceutical support.

6. In the online submission form, you indicated that Fully de-identified quantitative data will be made publicly available through the HPTN Dataverse on the Harvard Dataverse prior to publication.

Full qualitative transcripts will not be made publicly available due to the sensitive nature of the information contained therein, and the inability to ensure that a full transcript is completely de-identified. While direct identifiers and location names were not included in transcripts when they were prepared, there are no standardized procedures and tools for de-identifying sensitive qualitative data and removing indirect identifiers. In addition, qualitative research participants were assured during the informed consent process that their identity would remain confidential. Upon request, approval to access these data may be provided through a Data Use Agreement by FHI 360. Data access requests may be submitted to: Betsy Tolley (Director, Behavioral, Epidemiological and Clinical Sciences; FHI 360), BTolley@fhi360.org or Marie Shoen (Research Associate; FHI 360), MShoen@fhi360.org.

7. When completing the data availability statement of the submission form, you indicated that you will make your data available on acceptance. We strongly recommend all authors decide on a data sharing plan before acceptance, as the process can be lengthy and hold up publication timelines. Please note that, though access restrictions are acceptable now, your entire data will need to be made freely accessible if your manuscript is accepted for publication. This policy applies to all data except where public deposition would breach compliance with the protocol approved by your research ethics board. If you are unable to adhere to our open data policy, please kindly revise your statement to explain your reasoning and we will seek the editor's input on an exemption. Please be assured that, once you have provided your new statement, the assessment of your exemption will not hold up the peer review process.

8. One of the noted authors is a group or consortium. In addition to naming the author group, please list the individual authors and affiliations within this group in the acknowledgments section of your manuscript. Please also indicate clearly a lead author for this group along with a contact email address.

Reviewers' comments:

Reviewer's Responses to Questions

**Comments to the Author**

1. Is the manuscript technically sound, and do the data support the conclusions?

Reviewer #1: Yes

Reviewer #2: Yes

Reviewer #3: Yes

2. Has the statistical analysis been performed appropriately and rigorously? 

Reviewer #1: N/A

Reviewer #2: Yes

Reviewer #3: N/A

3. Have the authors made all data underlying the findings in their manuscript fully available?

Reviewer #1: Yes

Reviewer #2: Yes

Reviewer #3: Yes

4. Is the manuscript presented in an intelligible fashion and written in standard English?

Reviewer #1: Yes

Reviewer #2: Yes

Reviewer #3: Yes

5. Review Comments to the Author

**Reviewer #1:** The acceptability and individual preference of PrEP modalities will become more and more important as more PrEP products become available and research shifts to implementation. This work is a very important contribution highlighting the importance of considering not only stigma, but also practicability, short and long term side effects as well as monetary considerations.

My only major comment is that I would like to see a bit more quantitative analysis. For each pattern that is highlighted in the "findings" section, you should indicate how many respondents were asked a similar question. Sometimes this information is given and sometimes it is not or it is given only vaguely (about one third). I feel that we get a good quantitative breakdown of concerns prior to receiving injections, but it would be good to see how that compares with what was found.

I think this is especially important as these IDI transcripts will not be made publicly available due to justifiable privacy concerns.

Very minor comments:

Background

Line 64 “The jury is still out” suggest rephrasing to more precise language. Perhaps “numerous demonstration projects and implementation studies show that PrEP adherence among women is typically not sufficient to achieve complete protection” and citing Marrazzo (2024) on this point.

Generally, seems like some of these authors have done more work on this research question that should be highlighted in the introduction. Velloza et al (10.1002/jia2.25463) specifically addressed reasons for non-adherence on HPTN082.

Findings

Line 196 “By far, women appreciated the ease and convenience of a long-acting formulation” relative to an oral formulation?

**Reviewer #2**: This is a well written and interesting mixed methods manuscript that is important contribution to the literature as long-acting antiretrovirals are implemented in southern Africa. I only have minor comments for the authors to consider.

Introduction

Would recommend that the authors not use the trade names for TDF/FTC and TAF/FTC through out the manuscript. TDF is defined on line 45, but TAF/FTC is not (line 48).

Line 64: Would consider revising “the jury is still out” to something more descriptive such as “it has not been full evaluated”.

Line 70: Would include the full enrolled cohort, and not “> 3,200”

Methods

The methods are clear and outline the qualitative and quantitative approach for this analysis.

Results

The results are well written and clear. The section on injection site pain was a little confusion, which seems to be secondary to the wording of the second sentence on lines 225-226 Perhaps change to “Some participants described the pain as easily tolerable”.

Discussion

The discussion is well written and clear.

**Reviewer #3: **This manuscript used survey and qualitative data from a four-country sub-study in the HPTN 084 study to examine oral and injectable PrEP acceptability and considerations for CAB access among African women.

One of the key findings is “Participants overwhelmingly preferred injections to pills, ap- preciating the ease, convenience, and privacy of a long-acting formulation. ” However, such a finding is not fully demonstrated in the FINDINGS section. Specifically, the only relevant description is simply one sentence “Regardless of study arm, most women in the sub-study pre- ferred injections to daily pills when asked to indicate their preference; they often explained that the injection would be harder to forget to take and meant fewer clinic visits. ” I would suggest to include more details (e.g., proportion of women that prefer injection, quotes on forgetting taking PrEP and clinic visits)to strengthen the statement.

Some other comments:

1. Abstract: what do you mean by “A study statistician identified participants’ assigned study arm”?

2. Are the CPs also representative of early, mid-trial, or later entry? It is not clear from the description in the manuscript.

3. The SCs received up to two serial interviews. When are those interviews taken? Line 126: “Qualitative analyses presented in this paper focus on the first and second IDIs, both of which were conducted prior to participants’ ability to switch or remain on their current active product, i.e., during the blinded and unblinded periods of the study.” Is this statement also correct for the SCs?

4. Line 291: “Less frequently (but more often than for injections), women listed an array 1

of side effects they attributed to oral pill use. ” What is the frequency for side effect for those randomized to the CAB arm? Can you add the number to Line 271?

5. Minor comments:

• Line 99: “from” should be removed?

• Line 239: “single” is used for this women instead of “not married” by others.

6. PLOS authors have the option to publish the peer review history of their article (what does this mean?). If published, this will include your full peer review and any attached files.

Reviewer #1: **Yes: **Mia Moore

Reviewer #2: No

Reviewer #3: No

---

## [Author Response · Author response to Decision Letter 0]

9 Jul 2024

Responses to Reviewer:

Reviewer #1: The acceptability and individual preference of PrEP modalities will become more and more important as more PrEP products become available and research shifts to implementation. This work is a very important contribution highlighting the importance of considering not only stigma, but also practicability, short and long term side effects as well as monetary considerations.

My only major comment is that I would like to see a bit more quantitative analysis. For each pattern that is highlighted in the "findings" section, you should indicate how many respondents were asked a similar question. Sometimes this information is given and sometimes it is not or it is given only vaguely (about one third). I feel that we get a good quantitative breakdown of concerns prior to receiving injections, but it would be good to see how that compares with what was found.

I think this is especially important as these IDI transcripts will not be made publicly available due to justifiable privacy concerns.

Response: In response to this and reviewer 3’s comments, we have inserted additional numbers into several sections of findings. Qualitative researchers have different opinions about how to represent themes numerically, since not all participants are asked the same questions, or spontaneously raise topics in a standard way. We have tried to substantiate important themes with some numbers in ( ) without detracting too much from the qualitative descriptions. 

Very minor comments:

Background

Line 64 “The jury is still out” suggest rephrasing to more precise language. Perhaps “numerous demonstration projects and implementation studies show that PrEP adherence among women is typically not sufficient to achieve complete protection” and citing Marrazzo (2024) on this point.

Generally, seems like some of these authors have done more work on this research question that should be highlighted in the introduction. Velloza et al (10.1002/jia2.25463) specifically addressed reasons for non-adherence on HPTN082.

Response: Thank you for this suggestion. We have modified the introduction to discuss varying adherence trajectories described by Marrazzo (2024) and the Celum and Velloza articles. See highlighted section below.

While some research suggests that women using products with known efficacy can and will be more adherent (16), a pooled analysis of 11 post-approval PrEP studies conducted in cisgender women identified varying patterns of adherence over time (17). Based on a trajectory modelling approach that utilized available adherence data (including objective measures such as dried blood spots or tenofovir concentration measures, or more subjective measures like pill counts or self-reports) from almost 3000 women, about 40% of women reported consistent use (19%) or consistently high use (22% reported 4-6 pills per week) of PrEP up to 96 weeks of use. An equal proportion of women (39%) across these studies showed decreasing adherence over time, while about 21% reported consistently low adherence (17). Several smaller demonstration and implementation studies shed further light on these adherence patterns. For example, in a recent retrospective cohort study of 262 female PrEP initiators in Kisumu, Kenya, continuation dropped off significantly, with only 12% documented as having refilled their PrEP pills after 6 months (18, 19). Similarly, in a PrEP implementation study (HPTN 082) among 451 young women from South Africa and Zimbabwe, the majority (55%) had lapses in PrEP refills over a period of i12 months and only 21-22% were identified as highly adherent at six months (20). Women randomized to the advanced adherence arm of the study who reported at least one risk factor during a study visit were also more likely to have detectable drug on board, leading researchers to theorize that some young women might try to align their PrEP use with seasons of perceived risk (21). These social barriers continue to exist for women and make daily product use challenging.

Findings

Line 196 “By far, women appreciated the ease and convenience of a long-acting formulation” relative to an oral formulation?

Response: Yes, thank you. We have clarified.

Reviewer #2: This is a well written and interesting mixed methods manuscript that is important contribution to the literature as long-acting antiretrovirals are implemented in southern Africa. I only have minor comments for the authors to consider.

Introduction

Would recommend that the authors not use the trade names for TDF/FTC and TAF/FTC through out the manuscript. TDF is defined on line 45, but TAF/FTC is not (line 48).

Response: We have added the acronym for TAF/FTC in line 48. I have also removed the word “truvada” except for initial introduction. 

Line 64: Would consider revising “the jury is still out” to something more descriptive such as “it has not been full evaluated”.

Response: Thank you for this suggestion. We have expanded this section per reviewer 1 suggestions and removed this phrase. (See above response.) 

Line 70: Would include the full enrolled cohort, and not “> 3,200”

Response: Thank you. We have edited to read “3224”. 

Methods

The methods are clear and outline the qualitative and quantitative approach for this analysis.

Response: Thank you. 

Results

The results are well written and clear. The section on injection site pain was a little confusion, which seems to be secondary to the wording of the second sentence on lines 225-226 Perhaps change to “Some participants described the pain as easily tolerable”.

Response: Thank you for pointing out the need for some clarification. We have revised as follows:

For example, only a few participants randomized to the CAB arm, but many more in the TDF/FTC arm (5 versus 20), suggested that injectable pain was easily tolerated….

In contrast, many women described moderate to significant levels of pain, with some differences by arm.

Discussion

The discussion is well written and clear.

Reviewer #3: This manuscript used survey and qualitative data from a four-country sub-study in the HPTN 084 study to examine oral and injectable PrEP acceptability and considerations for CAB access among African women. 

One of the key findings is “Participants overwhelmingly preferred injections to pills, ap- preciating the ease, convenience, and privacy of a long-acting formulation. ” However, such a finding is not fully demonstrated in the FINDINGS section. Specifically, the only relevant description is simply one sentence “Regardless of study arm, most women in the sub-study pre- ferred injections to daily pills when asked to indicate their preference; they often explained that the injection would be harder to forget to take and meant fewer clinic visits. ” I would suggest to include more details (e.g., proportion of women that prefer injection, quotes on forgetting taking PrEP and clinic visits) to strengthen the statement.

Response: We have added proportion of women who clearly preferred CAB to oral PrEP. (A number of women indicated that either product would be good.) We didn’t include additional quotes because similar quotes can be found in other sections. 

Some other comments:

1. Abstract: what do you mean by “A study statistician identified participants’ assigned study arm”?

Response: The HPTN uses SCHARP – a statistical center in Seattle to conduct analyses. In order to stay blinded to study arm while conducting analysis, we requested that a study statistician from SCHARP draw up the lists of PTIDs, ensuring some balance across sites and arms, for the local study teams to then invite to participate. 

2. Are the CPs also representative of early, mid-trial, or later entry? It is not clear from the description in the manuscript.

Response: the CP IDs were drawn from across these different time points. However, we did not formally try to analyze data by study period, since the comparisons would consist of very small numbers. 

3. The SCs received up to two serial interviews. When are those interviews taken? Line 126: “Qualitative analyses presented in this paper focus on the first and second IDIs, both of which were conducted prior to participants’ ability to switch or remain on their current active product, i.e., during the blinded and unblinded periods of the study.” Is this statement also correct for the SCs?

Response: Thank you for this clarifying question. The statement is only applicable to the CPs, who took part in up to 3 different interviews. The final interview took place after participants were able to choose – and thus able to change the product they were using. Findings from those 3rd interviews will be presented in a subsequent paper. I have clarified that the first two IDIs of both CPs and SCs were included in this analysis.

4. Line 291: “Less frequently (but more often than for injections), women listed an array 1

of side effects they attributed to oral pill use. ” What is the frequency for side effect for those randomized to the CAB arm? Can you add the number to Line 271?

Response: Thank you for this comment. We have indicated the proportion of side effects for injectables in the section above (10 of 76) and amended the oral product experiences as below:

Additionally, more than half of participants (41 of 76) described encountering some problems with study pill adherence.

5. Minor comments:

• Line 99: “from” should be removed?

Response: No, I believe “from” should stay. We created a list and then selected PTIDs from the list. 

• Line 239: “single” is used for this women instead of “not married” by others.

Response: Thanks for pointing out this inconsistency. Now revised.

---

## [Decision Letter · Decision Letter 1]

20 Aug 2024

Preferences for Oral and Injectable PrEP among qualitative sub-study participants in HPTN 084

PONE-D-23-42740R1

Dear Dr. Tolley,

We’re pleased to inform you that your manuscript has been judged scientifically suitable for publication and will be formally accepted for publication once it meets all outstanding technical requirements.

Kind regards,

Tinashe Mudzviti, MPhil(MD)

Academic Editor

PLOS ONE

---

## [Editor Report · Acceptance letter]

26 Aug 2024

PONE-D-23-42740R1 

PLOS ONE

Dear Dr. Tolley, 

I'm pleased to inform you that your manuscript has been deemed suitable for publication in PLOS ONE. Congratulations! Your manuscript is now being handed over to our production team.

Kind regards, 

on behalf of

Dr. Tinashe Mudzviti 

Academic Editor

PLOS ONE